# miR-21-3p/IL-22 Axes Are Major Drivers of Psoriasis Pathogenesis by Modulating Keratinocytes Proliferation-Survival Balance and Inflammatory Response

**DOI:** 10.3390/cells10102547

**Published:** 2021-09-26

**Authors:** Florence Abdallah, Elodie Henriet, Amandine Suet, Ali Arar, Rudy Clemençon, Jean-Marc Malinge, Gaël Lecellier, Patrick Baril, Chantal Pichon

**Affiliations:** 1Centre de Biophysique Moléculaire, CNRS-UPR4301, 45071 Orléans, France; elodiehenriet@gmail.com (E.H.); amandinesuet@gmail.com (A.S.); Rudy.CLEMENCON@cnrs.fr (R.C.); jean-marc.malinge@cnrs.fr (J.-M.M.); patrick.baril@cnrs.fr (P.B.); 2Centre Hospitalier d’Orléans, 45071 Orléans, France; ali.arar@chr-orleans.fr; 3ENTROPIE, UMR250/9220, CEDEX, 98800 Noumea, New Caledonia, France; gael.lecellier@uvsq.fr; 4Laboratoire D’excellence “CORAIL”, USR 3278 CNRS-EPHE, Centre de Recherches Insulaires et Observatoire de l’Environnement, Papetoai, Moorea, Polynésie Française, Université de Paris-Saclay UVSQ, 55 Avenue de Paris, 78000 Versailles, France; 5Faculté des Sciences, Université d’Orléans, Colléguim Sciences et Techniques, 45071 Orléans, France

**Keywords:** IL-22, miR-21-5p, miR-21-3p, keratinocytes, psoriasis, proliferation

## Abstract

Psoriasis is a chronic inflammatory skin disease that is mediated by complex crosstalk between immune cells and keratinocytes (KCs). Emerging studies have showed a specific psoriatic microRNAs signature, in which miR-21 is one of the most upregulated and dynamic miRNAs. In this study, we focused our investigations on the passenger miR-21-3p strand, which is poorly studied in skin and in psoriasis pathogenesis. Here, we showed the upregulation of miR-21-3p in an IMQ-induced psoriasiform mouse model. This upregulation was correlated with IL-22 expression and functionality, both in vitro and in vivo, and it occurred via STAT3 and NF-κB signaling. We identified a network of differentially expressed genes involved in abnormal proliferation control and immune regulatory genes implicated in the molecular pathogenesis of psoriasis in response to miR-21-3p overexpression in KCs. These results were confirmed by functional assays that validated the proliferative potential of miR-21-3p. All these findings highlight the importance of miR-21-3p, an underestimated miRNA, in psoriasis and provide novel molecular targets for therapeutic purposes.

## 1. Introduction

Psoriasis is a chronic inflammatory skin disease caused by local and systemic activation of innate and adaptive immunity. It affects people of all ages, with an estimated 3% prevalence worldwide [1]. Psoriatic plaques manifest as thick red irritated skin lesions caused by imbalanced keratinocytes (KCs), hyperproliferation and differentiation impairment, as a result of intricate and overlapping immune pathways [2]. This epidermal hyperplasia—a key feature of psoriatic lesions—is due to an enriched inflammatory environment, in which IL-22, an IL-10 family cytokine, is highly upregulated [3]. IL-22 is mainly produced by T_H_22 and T_H_17 [4]. It acts on epithelial cells that exclusively express its heterodimeric receptor (IL-10RB/IL-22RA1), resulting in STAT3 downstream-dependent activation [5,6]. IL-22 induces cytokine-specific chemokines and specific effector responses mediated by the IL-23/T_H_17 axis [7]. Together with other inflammatory mediators, IL-22 stimulates KCs to produce inflammatory cytokines (IL-1β, TNF-α, IL-6), chemokines (CCL20), and antimicrobial peptides (S100A7, S100A8, S100A9, β-defensin-2), resulting in their hyperproliferation. In addition to IL-22’s pleiotropic functions, IL-22-deficient mice were shown to be almost protected from imiquimod (IMQ)-induced psoriasis, proving the key role of IL-22 in the pathogenesis of psoriasis [8]. These observations were recently consolidated in the psoriatic epidermal transcriptome, which identifies a dominated IL-22/IL-17 signature in KCs [9]. Moreover, a few years ago, a deregulated profile of MicroRNAs (miRs or miRNAs) in psoriasis was identified [10,11,12]. Briefly, miRNAs are small non-coding RNAs (~22 Nt) that function as key modulators of gene expression through the endogenous RNA interference machinery, resulting in translational repression or degradation of the target mRNAs. The multi-step miRNA biogenesis process yields two mirror strands, 5p and 3p [13,14]. Conventionally, the leader strand (5p) has been considered the least thermodynamically stable strand and thought to be the most effective miRNA. In line with these observations, miR-21-5p has been extensively studied. In skin biology, in addition to its oncogenic role in skin squamous cell carcinomas, miR-21-5p was shown to be essential for cutaneous wound healing. It acts by targeting KC proliferation, inflammation, collagen deposition, and melanogenesis by regulating melanin production via Sox5 inhibition [15,16,17,18,19,20]. Moreover, miR-21-5p was described to promote psoriatic skin inflammation by suppressing T cell apoptosis and by targeting TIMP3, which results in TACE activation [21,22]. Accordingly, in vivo miR-21-5p inhibition by antagomiR was shown to improve the psoriasis phenotype [22]. Very few studies have addressed the role of miR-21-3p, which is considered to be the passenger strand, in cutaneous systems, despites its clear upregulation in psoriatic skin and its implication in UV-induced skin inflammation [23]. As the stable passenger strand (3p or *), degradation was thought to happen immediately after genesis. However, several reports have demonstrated the co-expression of both leader and passenger strands, as well as their functionality in various biological systems [24]. Therefore, in this study, we aimed to characterize the biological activities of miR-21-3p in KCs under conditions of homeostasis and psoriasis pathology.

Here, we identified IL-22 as a key driver of miR-21-3p induction in the IMQ-induced psoriasiform skin inflammation model. We highlighted the diverse functions of miR-21-3p in interfering with KC proliferation, migration, inflammation and survival through RNA-seq analysis. Finally, we have proposed a landscape of overlapping pathways between a recently reported psoriatic epidermal transcriptome and KCs overexpressing miR-21-3p. Overall, our results point towards the implication of the passenger strand miR-21-3p in the onset of psoriasis in an IL-22-dependent manner. Hence, considering the miR-21-3p/IL-22 axis could open a novel therapeutic track for the treatment of psoriasis.

## 2. Materials and Methods

### 2.1. Mice and In Vivo Treatments

Wild-type (C57BL6/JRj) mice used in this study were purchased from Janvier Labs (Saint Berthevin, France). IL-22-deficient mice (IL-22^−/−^), kindly provided by Dr. Laure Dumoutier from Ludwig Institute for Cancer Research, were bred in the animal facility of the Centre Biophysique Moleculaire UPR4301 (Orléans, France). The experiments were performed in compliance with institutional guidelines and were approved by the local ethical committee. Mice between 8 and 10 weeks of age were shaved on the back skin and received daily, for 6 consecutive days, topical applications of 60 mg imiquimod (IMQ) from a commercially available cream (5%) (Aldara Hoofddorp, Pays-Bas; 3M Pharmaceuticals). Recombinant mouse IL-22 protein (R&D systems, #582-ML-010) at 100 ng was subcutaneously injected on the shaved back 1 day before IMQ treatment and throughout the rest of the experiment.

### 2.2. Scoring of Skin Inflammation Severity

To evaluate the severity of back skin inflammation, a scoring system inspired by the literature was developed. Erythema and scaling were scored independently on a scale from 0 to 4—0, none; 1, slight; 2, moderate; 3, marked; 4, very marked. The sum of erythema and scaling scores were established. Then, a score for each group was averaged, expressed in arbitrary units (a.u.) and trend lines were generated to observe the changes in mouse skin lesions.

### 2.3. Patients and Biopsies

For this study, 4-mm punch biopsies were collected from the lesional skin of 9 patients (5 women and 3 men, with an age range 18 to 60 years) with psoriasis at the Centre Hospitalier Régional d’Orléans (CHRO). Furthermore, 6-mm punch biopsies, collected from 9 healthy donors (9 women with an age range of 41 to 60 years), were supplied from the Laboratoire BIO-EC (Longumeau, France).

### 2.4. Cell Handling

Neonatal normal human epidermal keratinocytes (NHEKs) were purchased from Lonza (Levallois-perret, France) (#00192907) and immortalized human keratinocytes (HaCaT cells) were purchased from AddexBio catalog No. T002000. HaCaT cells were grown Dulbecco’s modified Eagle’s medium (DMEM), with 4.5 g/L Glucose, with l-glutamine (Lonza, catalog #BE12-604F) supplemented with 10% fetal bovine serum, and 1% penicillin (10,000 Units)—streptomycin (10 mg) (Sigma-Aldrich, (St. quentin fallavier, France), catalog # P0781) at 37 °C with 5% CO_2_.

### 2.5. Cell Stimulation

HaCaT cells were seeded at 2.5 × 10^5^ cells per well of 24-well plate or 6 × 10^5^ cells per well of 6-well plates for 24 h to obtain between 50% and 60% confluency. One day prior to stimulation, HaCaT cells were starved by replacing the growth medium with DMEM, with 4.5 g/L Glucose, with l-glutamine (Lonza, catalog # BE12-604F). Then, the stimulation was achieved in the same medium for 24 h with 50 ng/mL recombinant human IL-22 (R&D systems, # 782-IL010).

### 2.6. Cell Transfection

The transfection experiments were performed on 50% confluent cells in 12-well plates. **STAT3/ NF-κB silencing by siRNAs.** Si-STAT3 (forward: CCCGUCAACAAAUUAAGAATT; reverse: UUCUUAAUUUGUUGACGGGTT) and si-p65 (forward: GCCCUAUCCCUUUACGUCATT; reverse: UGACGUAAAGGGAUAGGGCTT); the siRNAs were transfected at a concentration of 30 nM using the lipofectamine RNAimax (#13778075, Invitrogen). **miR-21 gain and loss of function.** OligomiRs used for the transfection, purchased from Ambion Life Technologies were transfected at a concentration of 10 nM using the lipofectamine RNAimax (#13778075, Invitrogen (Villebon sur Yvette, France)). For inhibitors, negative control (#4464078), miR-21-5p (MH10206), and miR-21-3p (MH12979) were used, and for mimics, negative control (#4464060), miR-21-5p (MC10206), and miR-21-3p (MC12979) were used. The transfection protocol followed the manufacturer’s instructions. Cells were stimulated 24 h after this process and harvested 48 h after transfection for gene expression analysis and protein assays.

### 2.7. Luciferase Reporter Assays

One day prior to transfection, HaCaT cells were seeded onto 24-well plates at 1.5 × 10^5^ cells per well. The next day, cells were at around 60% of confluence and thus transfection was performed using the PTG1 transfection reagent (Polytheragene SAS, Evry, France) with 2.5 μg pNF-CMV-luc reporter vector [25] at a DNA/Polymer weight ratio of 1/6. Four hours after transfection, the medium was replaced with DMEM–high glucose supplemented with 1% FBS overnight. Then, the cells were treated as mentioned previously. Firefly luciferase gene expression was measured using luciferin (beetle luciferin Promega (#E1601)). Luciferase activity was normalized to total protein amount using a BCA protein assay kit (Interchim (Montluçon, France) protein quantitation kit, BCAssays, UP40840A).

### 2.8. RNA Extraction, Reverse Transcription, and Real-Time PCR

Total RNA and microRNA were isolated from the same piece of skin or from HaCaT cells using a mirVana™ miRNA Isolation Kit (Ambion Life Technology, (Evry, France) #AM1560), following the manufacturer’s instructions. Reverse transcription was performed on 1 μg total RNA with a RevertAid First Strand cDNA Synthesis Kit (Thermo Scientific (strasbourg, France), #K1622) and on 2 μL microRNA with a TaqMan™ Advanced miRNA cDNA Synthesis Kit (Applied biosystems (Villebon sur Yvette, France), #A28007). Quantitative PCR (qPCR) amplifications were performed using cDNA corresponding to 100 ng total RNA using primer sets suitable for Syber green detection using the Luna^®^ Universal Probe qPCR Master Mix (Promega, #M3004S). On the other hand, a 1/10 dilution of the amplified cDNA of miRNA were achieved using primer sets specific for each miRNA (TaqMan Advanced miRNA Assays, Applied Biosystems, A#25576) and the TaqMan Fast Advanced Master Mix (ThermoFisher Scientific (strasbourg, France), #4444557). The relative expression of CyclinD1 (forward: CCCTCACACCAATCTCCTCA; reverse: TCTTCTTCAAGGGCTCCAGG) was measured using the comparative critical threshold (CT) method, normalized to housekeeping gene β-actin (forward: GGGCGCCCCAGGCACCAGGT and reverse: CGTGCTCGATGGGGTACTTC) for total RNA and miR-16-5p (stable expression in cell lines and skin models used under the different conditions) for miRNA and determined using the formula 2^−ΔΔCT^. Primers for TaqMan MicroRNA assays were purchased from Thermo Fisher Scientific (Table 1).

### 2.9. Enzyme-Linked Immunosorbent Assay

To evaluate the inflammatory response, the culture medium was collected from treated keratinocytes and stored at −80 °C. The protein levels of IL-22, TNF-α, IL-1β, IL-4, and IL-6 were assayed using a DuoSet ELISA kit (R&D systems) according to the manufacturer’s instructions. To evaluate the inflammatory response in the psoriasis model, the same amount of skin protein lysates from biopsies was used. The concentration of proteins was evaluated using a BCA protein assay kit (Interchim (Montluçon, France) protein quantitation kit, BCAssays, UP40840A).

### 2.10. Phospho-STAT3 (Y705) Assay

The HaCaT cells were seeded into 96-well dishes at a density of 1 × 10^4^ cell per well, treated with 0 or 50 ng/mL IL-22 in triplicate for 24 h, and then lysed in the plate and assayed for total or pStat3 using the cell-based Stat3 ELISA kit according to the manufacturer’s instructions (R&D Systems, #DYC4607B).

### 2.11. Immunofluorescence Staining

The HaCaT cells were fixed with 4% paraformaldehyde for 15 min at room temperature. For Ki67 staining, cells were permeabilized with 0.1% triton X-100 for 5 min at room temperature. Then, cells were incubated with 1% BSA/0.3M glycine in 0.1% PBS-Tween for 1 h to reduce non-specific protein–protein interactions. Afterwards, cells were then incubated with anti-ki67 (ab15580) at 1 μg/mL concentration overnight at 4 °C; followed by a further incubation at room temperature for 1 h with a goat anti-rabbit IgG Alexa Fluor^®^ 488 at (Invitrogen A11034) at 4 μg/mL. Finally, the cells were mounted with VECTASHIELD Antifade Mounting Medium with DAPI used to stain the cell nuclei in blue (vector, H-1200). Image analysis was performed using FLoid^®^ Cell Imaging Station microscopy.

### 2.12. Crystal Violet Staining

The cells were fixed with ice-cold 100% methanol for 10 min, and then stained with 0.5% crystal violet solution in 25% methanol for 15 min at room temperature. Finally, the cells were washed in water several times, until the dye stopped coming off.

### 2.13. Cell Metabolic Activity

A Cell Proliferation kit II (XTT- Roche) allowed us to evaluate the cell metabolic activity by assessing the activity of intracellular oxydoreductases. XTT is a colorless or slightly yellow compound that becomes brightly orange upon reduction by cellular effectors as mitochondrial oxidoreductases. HaCaT cells were seeded in 24-well plates (2.5 × 10^5^ per well) and treated as indicated. An XTT assay was performed according to the manufacturer’s instructions. Briefly, the XTT solution at a final concentration of 0.3 mg/mL was added into each well and the incubation was carried out for 4 h at 37 °C. During this incubation, an orange formazan solution was formed, resulting from the conversion of the yellow tetrazolium salt XTT by viable cells. The intensity of coloration was quantified by measuring the absorbance with a specific absorbance filter at 450 nm using a Victor spectrophotomer (PerkinElmer, Waltham, MA, USA).

### 2.14. Flow Cytometric Analysis for Ki67 and PI Staining

HaCaT cells were harvested and spun down to a pellet at 1500 rpm for 5 min. The cells were washed with 1 mL of phosphate buffered saline (PBS-1X) (Sigma-Aldrich) and centrifuged at 1500 rpm for 5 min. The cells were fixed with 4% *p*-formadehyde at room temperature for 30 min. Then, the cells were pelleted by centrifugation, resuspended with a permeabilization buffer (50 µL/mL PI, 100 µg/mL RNase, and 2 mM MgCl_2_) (50 µL of buffer per 1 × 10^6^ cells). Afterwards, cells were centrifuged and stained with 5 µL of Alexa Fluor 700 mouse anti-human Ki-67 antibody (BD Pharmingen, #561277) and isotype control (BD Pharmingen, #557882), followed by room temperature incubation for 30 min, protected from light. Finally, the cells were washed twice with 2 mL of permeabilization buffer, resuspended, and analyzed with a BD LSR Fortessa Flow Cytometer (BD Biosciences).

### 2.15. Monitoring miRNA Activity with RILES

To monitor the functionality of miR-21-5p, -21-3p, -486-5p, and -486-3p in HaCaT cells, four RILES expression plasmids—denoted as pRILES/21.5pT, pRILES/21.3pT, pRILES/486.5pT and pRILES/486.3pT—were constructed as previously described [26]. Briefly, 4 fully complementary block sequences specific to either miR-21-5p, -21-3p, -486-5p, or -486-3p were subcloned in the 3′-UTR part of the CymR transcriptional gene. A control RILES plasmid, denoted pRILES, that did not contain any targeting sequence to any miR, was included in the study. To evaluate the functionality of endogenously expressed miR in HaCaT cells, the cells were seeded onto 24-well plates at 3.5 × 10^5^ cells per well. The following day, cells were transfected with 2.5 µg of RILES plasmids using PTG1 (Polytheragene SAS, Evry, France), at the pDNA/Polymer weight ratio of 1/4. After 4 h of transfection, the medium was replaced with DMEM-high glucose media supplemented with 0.5% serum overnight. Then, the cells were treated with IL-22 as described above. Firefly luciferase gene expression in cells was thereafter monitored 24 h later. Luciferase activity was normalized to total protein content using the BCA protein assay kit (Interchim, protein quantitation kit, BCAssays, UP40840A).

### 2.16. RNA-Seq from HaCaT Cells

Total RNA was extracted from the 3 replicates of HaCaT overexpressing 1 nM miR-21-3p mimic or negative control mimic using a mirVana™ miRNA Isolation Kit (Ambion Life Technology, #AM1560), following the manufacturer’s instructions. Small RNA libraries were prepared using 0.5 ng of total RNA per sample. The RNA-seq assay was carried out on an Illumina NextSeq 500 sequencer. Sequence reads in the FASTQ format were processed and the differential expression analysis was performed. This work has benefited from the facilities and expertise of the high-throughput sequencing core facility of I2BC (Centre de Recherche de Gif—http://www.i2bc.paris-saclay.fr/, accessed on 22 July 2020).

### 2.17. Statistical Analysis

Unless mentioned otherwise, all the experiments in the study were reproduced in at least two independent different experiments with a minimum of triplicates performed per group. All data are presented as mean ± SEM. The two-tailed Student *t*-test was used to determine significance between two groups. Ordinary two-way ANOVA was used for multiple comparisons of more than two groups. The simple linear regression model was used to analyze the correlation between RNA levels of inflammatory mediators and miR-21-5p or miR-21-3p. Comparisons between multiple groups were submitted to Bonferroni’s test and comparisons between 2 groups were submitted to Tukey’s test. *p*-values less than 0.05 were considered significantly different. * *p* < 0.05, ** *p* < 0.01, *** *p* < 0.001.

## 3. Results

### 3.1. miR-21-5p/3p Expression in IMQ-Induced Psoriasis in WT versus IL-22^−/−^ Mice

As a first approach, we compared the relative expression of miR-21 between WT and IL-22^−/−^ mice, which are known to develop a less severe form of the pathology [8] (Figure 1). We evaluated the clinical progression of the pathology upon IMQ application in WT and IL-22^−/−^ mice (Figure 1A and Appendix A). The application of IMQ at 60 mg on the shaved back skin of WT mice for six consecutive days caused erythema, scaliness, thickening, and acute inflammation (Figure 1A and Appendix A). In the early onset period (day 0 to day 4), the skin developed red erythematous plaques. Then, it turned into thickened and scaly skin (day 5 to day 7). A similar application on IL-22^−/−^ mice showed a less severe onset of clinical psoriatic lesions (Figure 1A and Appendix A).

Next, we analyzed the relative expression of miR-21 between the different groups of mice. Similar relative expression levels of both miR-21 strands were obtained for WT and IL-22^−/−^ healthy mice. At day 7 post-IMQ application, the expression level of miR-21 was significantly increased for both miR-21 strands (5p and 3p) in WT psoriatic mice compared to WT healthy mice. An upregulation of 4.3 times and 6.3 times for miR-21-5p and miR-21-3p respectively with respect to healthy mice was recorded (Figure 1B,C). In contrast to WT mice, IL-22^−/−^ psoriatic mice showed a slight increase of miR-21 expression (around 3-fold) compared to IL-22^−/−^ healthy mice for both miR-21 strands. Only the increased expression of miR-21-5p was statistically significant. Remarkably, WT psoriatic mice presented much higher miR-21 expression than IL-22^−/−^ psoriatic mice. This significant upregulation reached 1.3-fold for miR-21-5p and 2.3-fold for miR-21-3p.

Finally, we established a correlation between miR-21-5p (Figure 1D) and miR-21-3p (Figure 1E) expression with IL-22 mRNA and protein levels in WT mice. A significant positive correlation was noted with IL-22 mRNA and protein for both strands. Nevertheless, miR-21-3p had a higher positive correlation (*p* < 0.0001/*p* = 0.0007) than miR-21-5p (*p* = 0.03/*p* = 0.0151) (Figure 1D,E). Thus, this result suggests a connection between miR-21-3p and the IL-22 axis in psoriasis.

### 3.2. Validation of miR-21-3p Dependency on IL-22 In Vitro, in Psoriatic Human Biopsies, and In Vivo

Since in skin, KCs are the main target of IL-22 stimulation via IL-22RA1 expression [6], we investigated a correlation between the expression of miR-21 and IL-22 activity by treating in vitro HaCaT cells and NHEK with 50 ng/mL IL-22 for 24 h (Figure 2). The IL-22 treatment resulted in 1.4- and 1.3-fold increases of miR-21-5p expression in HaCaT cells and NHEK, respectively. A higher upregulation of miR-21-3p expression was obtained in both cell types with 2.5-fold increase in HaCaT cells and 2.2-fold increase in NHEKs (Figure 2A,B). In line with the above observations, miR-21 expression showed a positive correlation with IL-22 mRNA in human skin biopsies. Interestingly, this correlation was only significant for miR-21-3p (*p* = 0.03) in comparison to miR-21-5p (*p* = 0.269) (Figure 2C,D).

These findings drove us to evaluate whether the restoration of IL-22 in the IL-22^−/−^ psoriatic mice would reverse the expression pattern of miR-21 described above. To do so, we treated IL-22^−/−^ mice with recombinant IL-22 protein and followed the miR-21 expression (Figure 3). One day before and during IMQ or vehicle application, mice received 100 ng of IL-22 or the equivalent amount of physiological serum subcutaneously on the shaved back (Figure 3A). The miR-21 expression for both strands was assessed on day 3 (early response) and day 6 (late response) post-IMQ/vehicle application (Figure 3B,C). The mice’s body weight fluctuations are shown in Appendix A. IL-22-treated mice and control mice displayed similar levels of miR-21 expression with comparable expression profiles at days 3 and 6 post-treatments. Mice treated with IMQ showed a slight upregulation that reached around 1.26- to 2.7-fold increases for miR-21-3p and 3.2- to 4-fold increases for miR-21-5p. These upregulations rose to ~5-fold in mice that received both IL-22 and IMQ treatments. These observations demonstrate a relationship between IL-22 and miR-21-3p and to a lower extent with miR-21-5p. Of note, IL-22 treatment alone was not sufficient to induce miR-21-3p expression, suggesting the implication of other inflammatory mediators. Therefore, we examined the possible correlation of key cytokines (IL-1β, TNF-α, IL-4, and IL-6) known to drive crosstalk between KCs and immune sentinels, which results in psoriasis development [27,28], with the induction of miR-21-5p or miR-21-3p expression (Appendix A). A positive correlation trend was obtained with IL-1β, whereas IL-6 and IL-4 showed a negative correlation trend. These results suggest the interference of other inflammatory circuits downstream of the cytokine activation pathway. Nevertheless, we focused on understanding the mechanism by which IL-22 induces miR-21-3p expression, as they were strongly correlated, compared to other cytokines (Figure 1D and Appendix A).

### 3.3. IL-22 Induces miR-21-3p via STAT3 and NF-κB Pathways

We first looked at the miR-21 promoter region, which contains three and four potential binding elements for STAT3 and NF-κB, respectively (Figure 4A) [29,30,31]. The latter are known to be downstream transcription factors to IL-22 signaling and have been reported to be upregulated in lesional psoriatic skin [6,32,33,34]. Accordingly, the stimulation with 50 ng/mL IL-22 of HaCaT cells expressing the luciferase gene under a NFκB-driven promoter resulted in NF-κB nuclear translocation, as reported based on luciferase activity (Figure 4B). The involvement of STAT3 activation was shown via increased STAT3 phosphorylation over the total STAT3 ratio (Figure 4C). These data were validated by functional assays by investigating IL-22-dependent miR-21 induction when STAT3, NF-κB, or both were knocked down by siRNAs (Figure 4D). When those transcription factors were knocked down individually, the effect on miR-21 expression upon IL-22 stimulation was mitigated. However, the double knockdown of STAT3 and NF-κB resulted in clear miR-21 repression, despite IL-22 stimulation (Figure 4D). Altogether, these experiments suggest that the treatment of HaCaT cells with IL-22 induces a cell-signaling event favoring the activation of NF-κB and STAT3, which in turn enhances the transcription of the miR-21 gene. Moreover, we established a correlation between miR-21/STAT3, STAT3/IL-22, and miR-21/IL-22 from psoriatic patients treated with biologics, including tofacitinib (GSE69967), brodalumab (GSE53552), and adalimumab and methotrexate (GSE85034). The results from GSE69967 (tofacitinib) showed a significant correlation for miR-21/IL-22 (*r* = 0.32), miR-21/STAT3 (*r* = 0.75), and IL-22/STAT3 (*r* = 0.42). Likewise, data from GSE53552 (brodalumab) were significantly correlated for miR-21/IL-22 (*r* = 0.32), miR-21/STAT3 (*r* = 0.53), and IL-22/STAT3 (*r* = 0.55). Patients receiving GSE85034 (adalimumab and methotrexate) displayed significant correlations for miR-21/STAT3 (*r* = 0.55) and IL-22/STAT3 (*r* = 0.25) (data not shown). These results support the dependency of miR-21 on the STAT3 pathway driven, by IL-22, in agreement with our results. Consequently, these data showed that miR-21 induction happens at downstream of the IL-22 pathway via STAT3 and NF-κB stimulation.

### 3.4. IL-22-Induced miR-21-3p in KCs Is Functional

To evaluate the potential functional activity of miR-21-3p in the regulation of gene expression, we employed a miR-ON functional monitoring system previously developed in our lab (Figure 5) [26]. The RILES system relies on an engineered genetic circuit in which the expression of the luciferase reporter gene is switched ON when a miRNA of interest is processed by the RISC machinery in cells to exercise its mRNA-functional repressive function (Figure 5A). The RILES system was engineered to bear miR-21-5p or miR-21-3p targeting sequences (pRILES/21.5pT and RILES/21.3pT, respectively). As shown in Figure 5B, the pRILES/21.5pT system was not significantly switched ON in HaCaT cells in response to several doses of IL-22 ranging from 0 to 50 ng/mL. In contrast, the RILES/21.3pT was significantly switched ON in HaCaT cells treated with the same conditions (Figure 5B). As an internal control, with an irrelevant miR target, we demonstrated that neither the pRILES/486 5pT- nor the pRILES/486 3pT-bearing targeting sequences of miR486-5p and miR486-3p were switched ON in response to IL-22 (Appendix A). Hence, our monitoring system showed specific IL-22/miR-21-3p induction. These observations reflect functional miR-21-3p activity in IL-22-stimulated HaCaT cells, reinforcing our data. Henceforward, we aimed at understanding the role of miR-21-3p in KC biology by defining its genetic regulatory network.

### 3.5. Global Changes in the Transcriptome of HaCaT Overexpressing miR-21-3p

Most studies have focused on determining one direct gene target per miRNA. In this study, we decided to have a global view of the gene network influenced by miR-21-3p upregulation. As a matter of fact, miRNA gene regulation is much more complex and could implicate several pathways that are interconnected, via the direct or indirect targeting of single or multiple gene sets. Therefore, we chose to use the RNA-sequencing approach to explore the transcriptomic landscape of miR-21-3p-overexpressing KCs. The overexpression of miR-21-3p was confirmed via RT-qPCR (Appendix A). The comparison between KCs overexpressing miR-21-3p versus the negative control (NC) revealed a significant (*p.adj* < 0.05) set of 660 downregulated genes and a set of 884 upregulated genes among the 22,152 detected genes (Figure 6A and Appendix A). These results indicate global changes (around 7%) in the transcriptome signature upon miR-21-3p overexpression. The impact of miR-21-3p upregulation on KCs’ molecular and biological pathways was analyzed via gene-annotation enrichment analysis to identify differentially expressed genes (DEGs) using ShinyGO v0.61 and DAVID 6.7 software. Both bioinformatic tools gave similar results. The top 30 biological processes of the downregulated genes, as provided by ShinyGO (Figure 6B), were mainly related to developmental/differentiation, locomotion/migration, cell communication, and immune system processes (Appendix A, ShinyGO). Consistently, the DAVID analysis highlighted similar biological processes, including the interferon-gamma-mediated signaling pathway (Appendix A, DAVID). On the other hand, almost all the upregulated genes in KCs were found in the categories related to immune responses and cellular proliferation, involving cell division, mitotic machinery, cell cycle transition, DNA replication, DNA repair, metabolic processes, RNA machinery, etc. (Figure 6C and Appendix A, DAVID and ShinyGO). It worth pointing out the enrichment for the NIK/NF-κB signaling pathway, as well as the T cell receptor signaling and response to cytokine IL-22-dependent pathways.

The proximal region (<600 bp) of these sets of DEGs was also analyzed and significant enriched motifs in promoters were found (Appendix A). Among the motifs that were significantly enriched in the downregulated gene set, the NR2C2 transcription factor gene, also known as TR4, and its motif were noted in this set. This nuclear receptor and the members of this family participate in many biological processes, such as development, cellular differentiation, and homeostasis [35,36]. Most of the significant motifs include transcription factor motifs implicated in differentiation processes, such as ZNF281 for embryonic stem cell differentiation; the TCF3 motif of the E protein family, which plays a critical role in lymphopoiesis, required for B and T lymphocyte development; and MZF1 for hemopoietic development [37,38]. One motif of NFKB1 was also included in this gene set [39,40].

For upregulated genes, there was an over-representation of motifs belonging to the EF2 promotor family (2 E2F1, 1 E2F2, 1 E2F4 and 1 E2F6), of which the E2F2 gene was augmented. The E2F family plays a crucial role in cell cycle control and tumor suppressor activity [41]. In the same way, KLF7 regulates cell proliferation, differentiation, and survival [42], as well as NRF1, regulating cellular growth and nuclear genes required for respiration, heme biosynthesis, and mitochondrial DNA transcription and replication [43,44]. Another transcription factor family that was over-represented was the CxxC family, which is implicated in chromatin organization and chromatin regulation/acetylation and in methylation patterns following DNA replication. There was also the HES4 motif, a transcriptional repressor implicated in the PI3K-Akt signaling pathway, and ELK4, related to the NGF pathway and ERK signaling.

To further evaluate whether the transcriptomic change observed in our study might correlate with a similar change in situ, in KCs derived from psoriatic lesions, we took advantage of the data recently published by the team of Sonkoly on the KC transcriptome in psoriasis, which showed an IL-22 enrichment, for a comparison with our data [9]. Thereby, we established a series of comparisons between differentially expressed genes in KCs overexpressing miR-21-3p versus psoriasis lesional skin (PP) or psoriasis non-lesional skin (PN) and healthy skin (H). If most of the DEGs in KCs overexpressing miR-21-3p are distinct (1149/1544) from those found in PP and PN, all DEGs in common (395/1544—25%) concern the DEGs of PP, of which 87 were shared strictly with PP vs. H and 303 were shared with PP vs. H and PP vs. PN (Figure 7A). The significantly enriched biological processes of these 395 genes concerned DNA replication and the cell cycle (Figure 7B). Moreover, regardless of the number of DEGs in common, the significantly enriched biological processes of all the upregulated genes in KCs overexpressing miR-21-3p and the top 10 genes in PP were similar (Figure 7A) [9]. These biological processes concerned DNA replication and repair, the cell cycle, and NIK/NF-kappaB signaling. Conversely, the development and differentiation processes of KCs were not shared. No significant similarly enriched biological processed (GO terms) were found with downregulated genes, even among those concerning differentiation and development (Table 2).

In the following part of this study, we intended to validate the occurrence miR-21-3p-dependent cellular processes, as revealed by our RNA-seq results. For that purpose, we next assessed the effect of miR-21-3p on KC proliferation and migration.

### 3.6. miR-21-3p Promotes Keratinocyte Hyperproliferation

Since miR-21-3p overexpression upregulates genes related to cellular survival and proliferation, as suggested above, and since KC hyperproliferation is a hallmark feature of psoriatic plaque formation, we examined the impact of miR-21-3p overexpression on KC proliferation. To this end, we overexpressed or inhibited miR-21-3p in KCs, using oligomers. The efficiency of miR-21 overexpression (Appendix A) and inhibition (Appendix A) was confirmed via RT-qPCR analysis. Then, we assessed the proliferation via a crystal violet assay, an XTT assay, and Ki67 and PI staining for the cell cycle (Figure 8 and Appendix A).

The crystal violet assay indicated that KC density depended on miR-21-5p and miR-21-3p expression status (Figure 8A and Appendix A). Indeed, kinetic studies indicated that KCs overexpressing miR-21-5p or miR-21-3p presented better growth compared to negative controls and to miR-21-5p or miR-21-3p inhibition over time. We also noted a more rapid increase in proliferation for miR-21-3p-expressing KCs at an earlier time (starting from 24 h) than KCs overexpressing miR-21-5p (Figure 8A). These observations were reinforced by the XTT results, which showed a highly significant amplified proliferation rate for miR-21-3p overexpression compared to negative controls (NC) and miR-21-5p mimics (Figure 8B). It is noteworthy that miR-21 inhibition resulted in a limited proliferation enhancement (Appendix A). Accordingly, we observed an increased expression of the proliferation marker Ki67 in miR-21-overexpressing KCs, whereas Ki67 was reduced when miR-21 was inhibited (Figure 8D). The outcome of miR-21 inhibition was modest compared to the efficiency (Appendix A). According to the literature, RT-qPCR data could incorrectly indicate a higher inhibition of miRNA due to masking effects [45]. Nevertheless, in our case, miR-21 inhibition gave encouraging results, in line with miRNA mimic data. In addition, cell cycle analysis revealed a significant 15% increase in S-phase-proliferating miR-21-3p-overexpressing KCs compared to NCs (Figure 8C). Consequently, this increase was absent upon miR-21-3p inhibition (Appendix A). Furthermore, the percentage of S-phase-proliferating cells compared to G0-G1 cells was 13% for miR-21-5p-overexpression in KCs, whereas it reached 30% for miR-21-3p-overexpressing KCs. Given all these results, we analyzed CylcinD1, an essential regulator of G1-S phase transition that promotes cell proliferation, at the mRNA level via RTqPCR analysis, and the proteomic level via Western blot analysis (Appendix A) [46]. Consistently, the overexpression of miR-21-3p and miR-21-5p resulted in 2.7-times and 1.7-times increases in CyclinD1 mRNA expression, respectively (Appendix A). At the protein level, the overexpression of both miR-21 strands caused CyclinD1 upregulation, with 33% and 39% for miR-21-5p and miR-21-3p mimics, respectively (Appendix A). Overall, these results support the RNAseq data, suggesting that miR-21-3p promotes the hyperproliferation of KCs.

## 4. Discussion

Despite all the findings and research advances in the biology miRNAs, skin miRNAs remain a relatively recent area of study, lacking understandings and mechanistic insights on the role and function of miRNAs in skin biology and disorders. In this study, we were interested in investigating miR-21, in particular the miR-21-3p strand, which is poorly studied in cutaneous systems. For this reason, we evaluated the expression of the two versions of miR-21 (5p and 3p) and showed their increased expression in the development of psoriasis. In the literature, only one study in addition to our results revealed the upregulation of miR-21-3p, considered as passenger strand, in psoriatic lesions, without further investigations of its role in psoriasis [23]. Interestingly, only few days ago, Li et al. showed an upregulation of miR-21-3p in NHEK co-cultured with psoriatic dermal mesenchymal stem cells, pointing to the importance of this passenger miRNA in psoriasis [47]. Although miR-21-3p expression was lower than that of miR-21-5p, our findings are in agreement with previous evidence suggesting that the passenger strand is usually less abundant than the mature miRNA strand [23,48]. Similar observations were noted in ovarian cancer and in UV-induced KC inflammation models [23,49]. However, for the first time, the current study demonstrates the co-expression of leader and passenger strands of miRNAs in skin, long thought to be subject to strand selection, which is believed to be dependent on the relative thermodynamic stability of the two ends of the duplex. The leader strand is considered the effective strand, whereas, until now, no clear and solid evidence has existed concerning the fate and the mechanism of removal or degradation of the passenger strand. Some studies have reported that this passenger strand might be destined either to be degraded as a mere carrier strand or to be expressed abundantly as a potential functional guide miRNA [48,50,51,52,53]. This study introduces new evidence on the potential functional expression of both strands, which has been underestimated.

Even though the etiology of psoriasis is still not fully understood, a complex crosstalk between KCs and the abnormal activation of the innate as well as adaptive immune responses that drive epidermal hyperplasia via a large panel of inflammatory mediators, is very well defined [27]. Therefore, we examined cytokines to search for a potential trigger of miR-21 using a correlation statistics test. We identified IL-22 as a major trigger of miR-21 in lesional skin, with a strong relationship with miR-21-3p. Furthermore, we confirmed these results in vitro using different KC cell types and human skin biopsies, as well as in vivo using an IL-22^−/−^ IMQ mouse model. Of note, we were expecting an upregulation of miR-21-3p expression in IL-22-treated IL-22^−/−^ mice, but these mice did not modulate its expression. This suggests the interference of other mediators that are not observed under steady-state conditions. Indeed, IL-1β and TNF-α were shown to stimulate miR-21-3p expression in human aortic endothelial cells [54]. Most probably, the inflammatory storm in psoriasis drives the activation of several inflammatory pathways that, in one way or another, promote miR-21-3p expression either by recruiting immune cells or by inducing the hyperproliferation of KCs. Nevertheless, we believe that the primary induction of miR-21-3p by IL-22 would initiate both KC hyperproliferation and an exacerbated inflammatory response, leading to the formation of a positive feedback loop, which may interfere with miR-21-3p transcription. However, the use of RILES technology allowed us to underline the functional activity of miR-21-3p in KCs in response to IL-22 stimulation, consequently putting this cytokine at the center of psoriasis immunopathology and miRNA modulation [55,56].

We also demonstrated that IL-22 induces miR-21-3p transcription directly or indirectly through STAT3 and NF-κB signaling, which are crucial mediators of several biological and immunological processes, including inflammation and proliferation, subsequently ensuring the connection between KCs and immunocytes in the pathogenesis of psoriasis [57,58]. These results suggest that the overexpression of miR-21 in psoriatic KCs is likely a consequence of the production of excess inflammatory cytokines, in particular IL-22 in skin lesions, and may be implicated in the occurrence of epidermal hyperplasia in psoriasis. Thus, in the forthcoming years, a global view of the psoriasis immunopathogenesis events is needed in order to understand the connections between all inflammatory actors, from cellular to molecular mediators, including miRNAs. We would like to address a curious difficulty in the detection of NF-κB signaling mediated by IL-22, as the time of detection is challenging. A possible explanation is that IL-22 upregulates miR-21-3p, which has been revealed to impair p65 nuclear translocation [54]. Hence, it is only at early stages of IL-22 stimulation that NF-κB induction detection would be possible.

In line with our hypothesis that miR-21-3p and the IL-22 axis play essential roles in psoriasis, we performed RNA-seq analysis in miR-21-3p-overexpressing KCs to gain deeper insights into the transcriptome. Our results highlight an overall upregulation in the categories related to immune responses and cellular proliferation and a downregulation of networks related to the interferon-gamma-mediated signaling pathway and differentiation processes. These observations on the proliferation–differentiation balance were confirmed by multiple functional assays that validated the capacity of miR-21-3p to boost KC proliferation and migration. A very interesting and intriguing outcome of miR-21-3p overexpression is the downregulation of IFN-γ signaling pathways. Undeniably, IFN-γ is a critical cytokine involved in several cellular processes and defenses including proliferation and innate-adaptive immunity against invading pathogens. In steady-state conditions, IFN-γ drives the growth arrest of KCs [59]. This is in line with the proliferative properties of miR-21-3p observed in our study. On the other hand, in psoriasis, IFN-γ is thought to mediate T cells’ interactions with KCs. Furthermore, IFN-γ was shown to correlate with disease severity and prognosis [60]. Incontrovertibly, the downregulation of IFN-γ signaling raises a fundamental question about the nature of miR-21-3p regulation in psoriasis and points to its cell-specific function. Hence, further investigations on the function of miR-21-3p in other cellular subsets would be required in order to understand its regulatory mechanism in inflammatory diseases. Additionally, miR-21-3p overexpression upregulated genes linked to non-canonical NF-κB and T cell receptor signaling. These findings point to a probable role of miR-21-3p in exacerbating the inflammatory response. In fact, NF-κB hyperactivation is one of the main factors causing inflammation in psoriasis. The upregulation of non-canonical NF-κB signaling is dependent on NIK (MAP3K), which would mediate a long and persistent activation of the p52 complex [61]. It is plausible that NIK promotes TCR-stimulated STAT3 phosphorylation, thereby regulating the polarization of T naïve cells into T_H_17 phenotypes and promoting the proliferation of KCs [62]. This could be a very interesting point to consider, as the IL-17/IL-23 axis plays an essential role in the pathogenesis of psoriasis [63,64]. IL-23 inhibitors revolutionized the management of psoriasis with their high efficacy and the fact that they were well-tolerated when used as treatments for moderate-to-severe-plaque psoriasis [63,65,66]. More recently, the strength of IL-17 has been proven with the use of the new generation of biologics against IL-17A and IL-17F. Indeed, patients with moderate-to-severe-plaque psoriasis treated with IL-17 inhibitors achieved a long-lasting PASI > 90 [67,68,69,70]. Altogether, the RNA-seq results provide accumulating evidence on the abnormal immune regulatory genes induced by miR-21-3p overexpression, which are involved in the molecular pathogenesis of psoriasis.

Finally, this study highlights the pleiotropic functions of miR-21-3p in the cutaneous system, from regulating proliferation and cell survival to potentially modulating innate and adaptive immunity. In light of these findings, the role of miR-21-3p should be further addressed in upcoming studies by covering the different cell subsets present in the skin, such as melanocytes, T cells, Langerhans cells, etc. Altogether, these results open novel therapeutic strategies in which miR-21 inhibition of both strands should be considered. Furthermore, one can imagine a combination of therapeutics, in which the naturally occurring IL-22 antagonist (IL-22BP) could serve as a complement to counterbalancing or limiting miR-21 induction, without inducing side effects as in the case of biologics.

## Figures and Tables

**Figure 1 cells-10-02547-f001:**
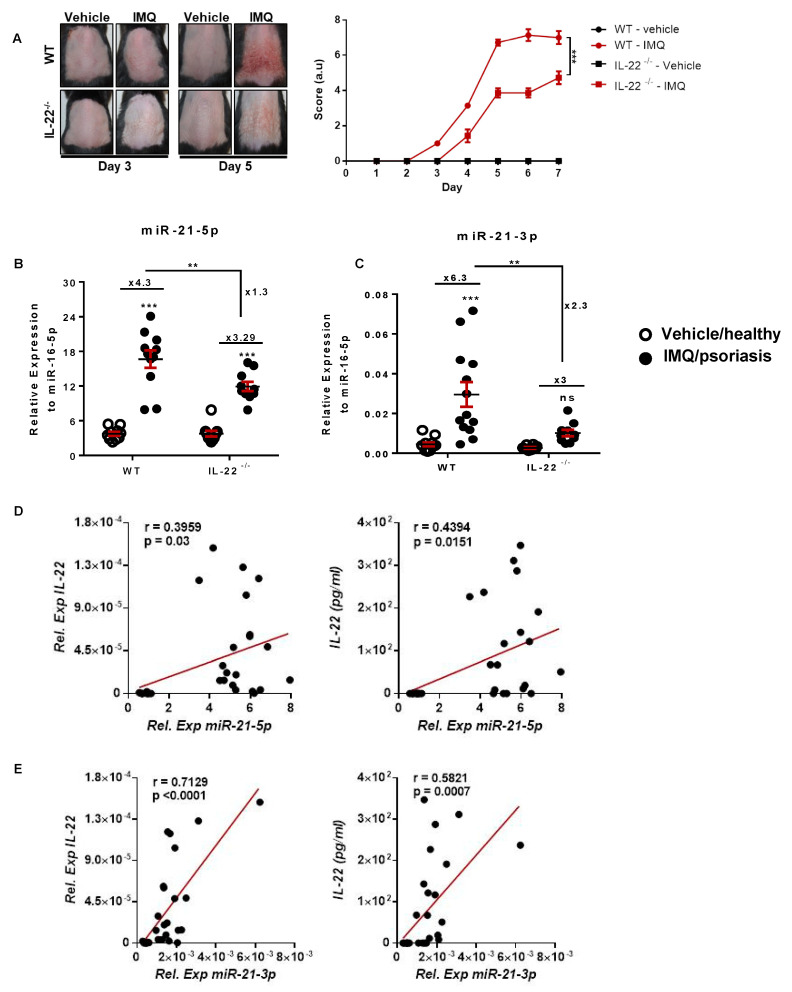
miR-21 (5p and 3p) expression patterns during psoriasis development. IMQ was applied daily for 6 consecutive days on the shaved backs of C57BL/6 wild-type (WT) and IL-22-knockout mice (IL-22^−/−)^. Pathogenesis was studied from day 1 to day 7. (**A**) Pictures of mice representing skin lesions throughout the development of psoriasis skin lesions (left), cumulative score representations (a.u.) of erythema and scaling, on the back skin, scored daily on a scale from 0 (absence of severity) to 4 (highest severity) (right). (**B**,**C**) RT-qPCR quantification of the relative expression of miR-21-5p- or miR-21-3p-to-miR-16-5p levels in lesional skin of mice treated with IMQ. (**D**) Pearson correlations established in 30 mice comparing the relative expression of miR-21-5p and IL-22 relative expression or IL-22 concentration (pg/mL) from skin protein lysates, assayed by ELISA. (**E**) Pearson correlations established in 30 mice comparing the relative expression of miR-21-3p and IL-22 relative expression or IL-22 concentration (pg/mL) from skin protein lysates, assayed by ELISA. Data information: results are presented as mean values ± SEM. Data are representative of two independent experiments with five mice per group. The statistical comparisons between groups were performed using the two-tailed Student *t*-test: ** *p* < 0.01, *** *p* < 0.001.

**Figure 2 cells-10-02547-f002:**
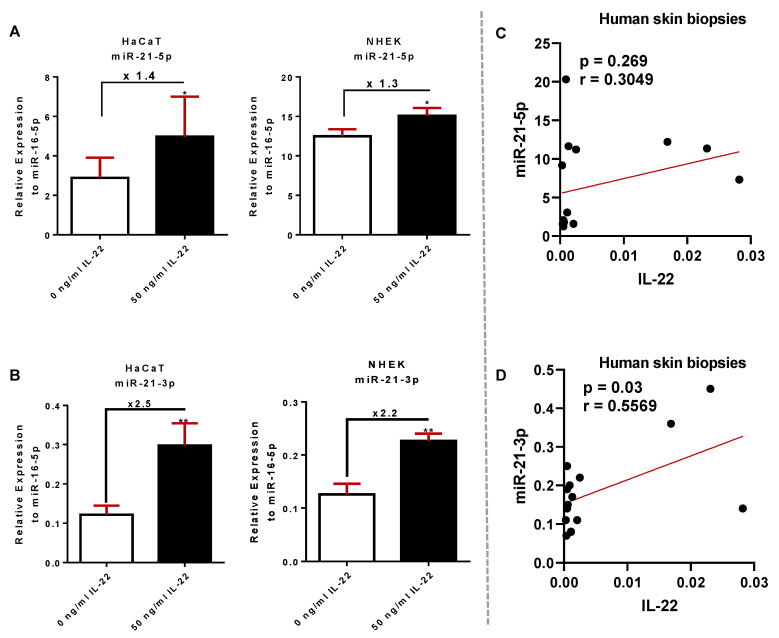
IL-22 correlates with miR-21-3p and to a minor extent with miR-21-5p induction. (**A**) RT-qPCR quantification on the relative expression of miR-21-5p compared to miR-16-5p levels in HaCaT cells and NHEKs stimulated with 50 ng/mL IL-22 for 24 h. (**B**) RT-qPCR quantification of the relative expression of miR-21-3p compared to miR-16-5p levels in HaCaT cells and NHEKs stimulated with 50 ng/mL IL-22 for 24 h. (**C**) Pearson correlations between relative the expression of miR-21-5p and IL-22 in human skin biopsies from healthy (*n* = 10) and psoriatic individuals (*n* = 9). (**D**) Pearson correlations between the relative expression of miR-21-3p and IL-22 in human skin biopsies from healthy (*n* = 10) and psoriatic individuals (*n* = 9). Data information: results are presenteds as mean values ± SEM. Data are representative of three independent experiments done in triplicates. The statistical comparison between groups was performed by using a two-tailed Student’s *t*-test: * *p* < 0.05, ** *p* < 0.01.

**Figure 3 cells-10-02547-f003:**
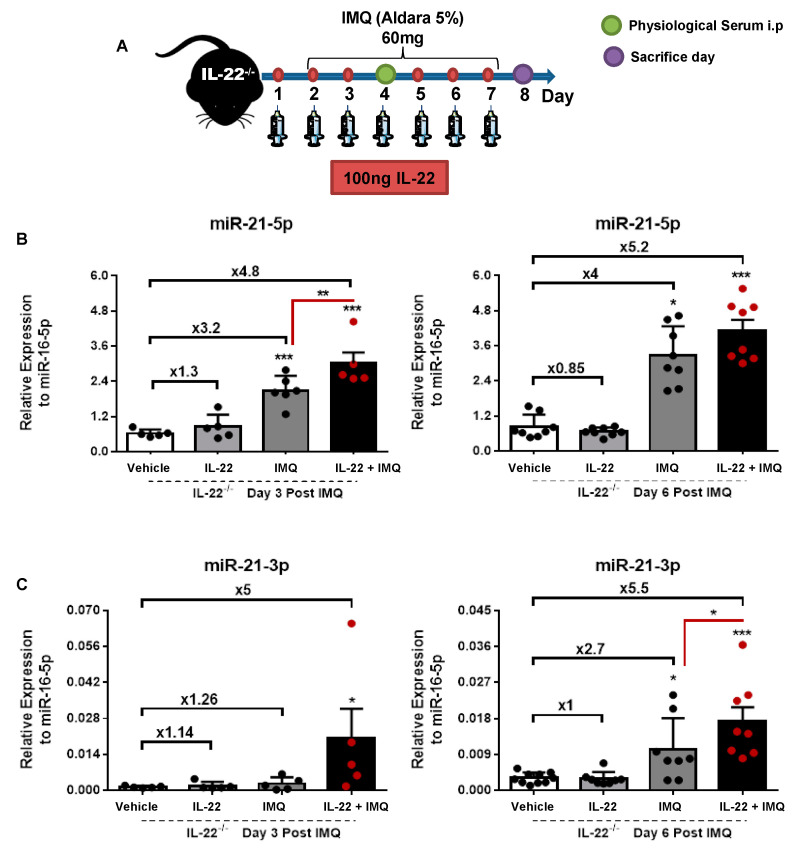
miR-21 induction in psoriasis is IL-22-dependent. (**A**) Schematic representation of IMQ-induced psoriasiform or control mice (vehicle application)—time course of IL-22^−/−^ mice that received 100 ng of IL-22 or an equivalent amount of physiological serum subcutaneously on the shaved area, one day before the induction of pathogenesis and up to day 7. (**B**) Relative expression of miR-21-5p to miR-16-5p at day 3 and day 6 post-IMQ application in vehicle mice (white histogram), IL-22-treated mice (light gray histogram), IMQ induced psoriasiform mice (dark gray histogram), and IMQ-induced psoriasiform mice treated with IL-22 (black histogram). (**C**) Relative expression of miR-21-3p to miR-16-5p at day 3 and day 6 post-IMQ application in vehicle mice (white histogram), IL-22-treated mice (light gray histogram), IMQ-induced psoriasiform mice (dark gray histogram), and IMQ-induced psoriasiform mice treated with IL-22 (black histogram). Data information: results are presented as mean values ± SEM. Data are representative of two independent experiments with at least five mice per group. The statistical comparison between groups was performed using two-way ANOVA: * *p* < 0.05, ** *p* < 0.01, *** *p* < 0.001.

**Figure 4 cells-10-02547-f004:**
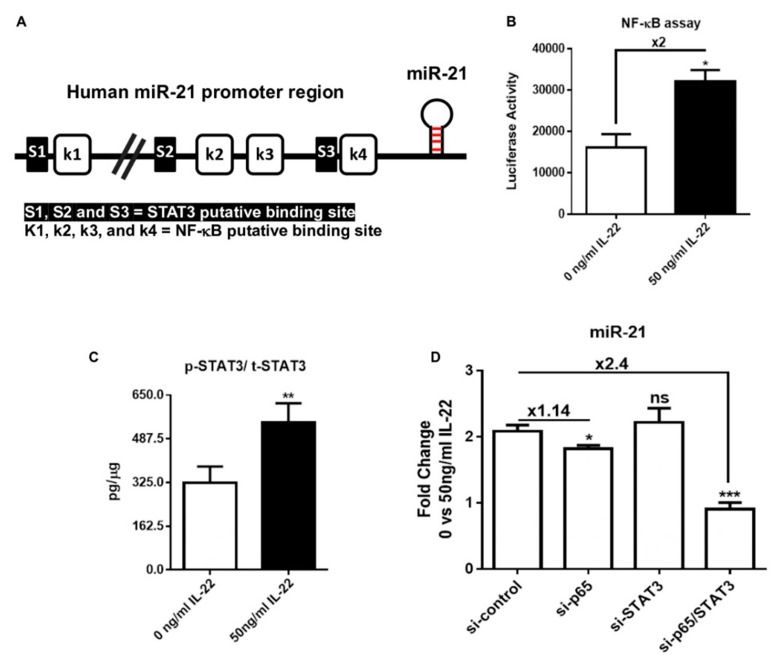
Requirements of STAT3 and NF-κB signaling for the induction of miR-21-3p in IL-22-stimulated keratinocytes. (**A**) Schematic representation of the miR-21 promoter gene with the putative binding sites for STAT3 (S1, S2, and S3) and NF-κB (k1, k2, k3, and k4) transcription factors. (**B**) The luciferase activity of NF-κB in HaCaT cells transfected with NF-κB luciferase reporter construct was measured after 24 h of 50 ng/mL IL-22 stimulation. (**C**) IL-22 induction of the STAT3 pathway in HaCaT cells was evaluated using a cell-based ELISA kit. (**D**) Representation of the fold change in miR-21 (5p and 3p) relative expression, assessed via RT-qPCR. The fold change corresponds to the ratio of 50 ng/mL IL-22 over 0 ng/mL IL-22-treated HaCaT cells and silenced with si-Ctrl, si-p65, si-STAT3, or si-p65/STAT3 for 24 h. Data information: results are presented as mean values ± SEM. Data for B and C are representative of two independent experiments and for D are representative of three independent experiments performed in triplicate. The statistical comparison between groups was performed using the two-tailed Student’s *t*-test: * *p* < 0.05, ** *p* < 0.01, *** *p* < 0.001.

**Figure 5 cells-10-02547-f005:**
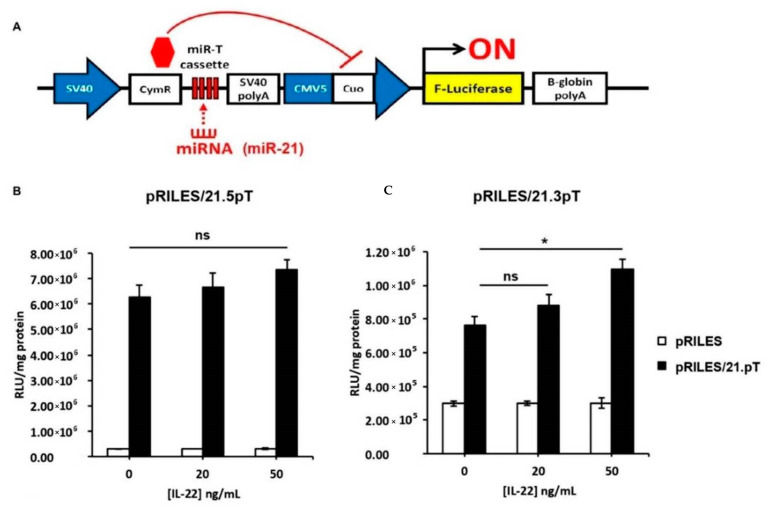
Functional assay using the RNAi-inducible luciferase expression system (RILES) to monitor miR-21 activity in HaCaT cells. (**A**) Schematic representation of the RILES construction. Briefly, the firefly luciferase reporter gene is repressed by a translational repressor (CymR—in red). The latter is under the control of the miR-21 cassette (miR-21.5pT or miR-21.3pT cassette, black histogram). Thus, the binding of miR-21-5p or miR-21-3p to the miR-21.5pT and miR-21.3pT cassette, respectively, results in CymR degradation that is translated by luciferase expression. Changes in luciferase levels reflect whether or not a miRNA can induce CymR mRNA degradation or translation arrest, which indicate miRNA functionality. An empty miR cassette, pRILES, was used as a control plasmid. (**B**,**C**) miR-21-5p and miR-21-3p expression induction assessments using RILES technology in HaCaT cells stimulated with 0, 20, 50, and 100 ng/mL of IL-22 for 24 h. Data are presented as relative luciferase units (RLU) normalized to the total amount of protein (mg). Data information: results are presented as mean values ± SEM. Data are representative of three independent experiments performed in triplicate. The statistical comparison between groups was performed by using two-tailed Student’s *t*-test: * *p* < 0.05.

**Figure 6 cells-10-02547-f006:**
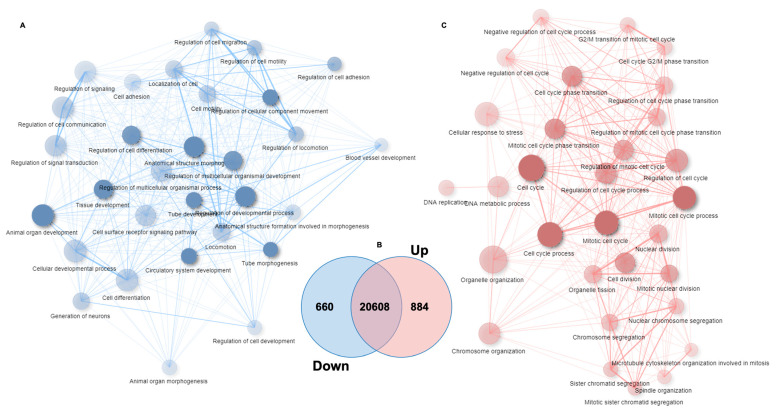
DEG analysis in KCs overexpressing miR-21-3p versus negative control. (**A**,**C**) Functionally grouped networks for significant downregulated genes (blue) and upregulated genes (red). Only the labels of the 30 most significant terms per group are shown. Each node represents an enriched GO term. Related GO terms are connected by a line, the thickness of which reflects the percentage of overlapping genes. The size of the node corresponds to the number of genes. The network was automatically laid out using the ShiniGo tool with an edge cut-off of 0.2. Data information: data are representative of experimental triplicates per condition. (**B**) Venn diagram. showing the extent of overlap among all expressed genes between the miR-21-3p overexpressing HaCaT cells versus NCs. The level of significant gene modulation was *p.adjusted* < 0.05.

**Figure 7 cells-10-02547-f007:**
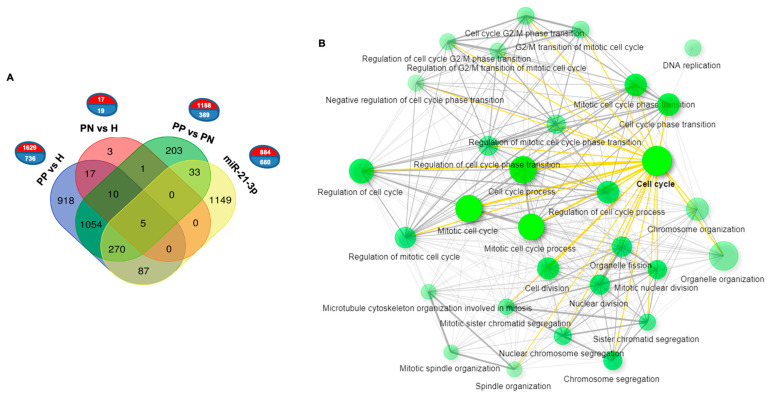
Comparative analysis between miR-21-3p-overexpressing KCs and psoriatic skin. (**A**) Graphs representing the numbers of significant DEGs (red: upregulated; blue: downregulated) between the indicated groups (PP vs. H, PN vs. H and PP vs. PN, from Pasquali et al., 2019 [9] and miR-21-3p, from this work). The Venn diagram shows the extent of overlap among DEGs. (**B**) Enrichment analysis of the 395 genes upregulated in PP keratinocytes and induced by miR-21-3p.

**Figure 8 cells-10-02547-f008:**
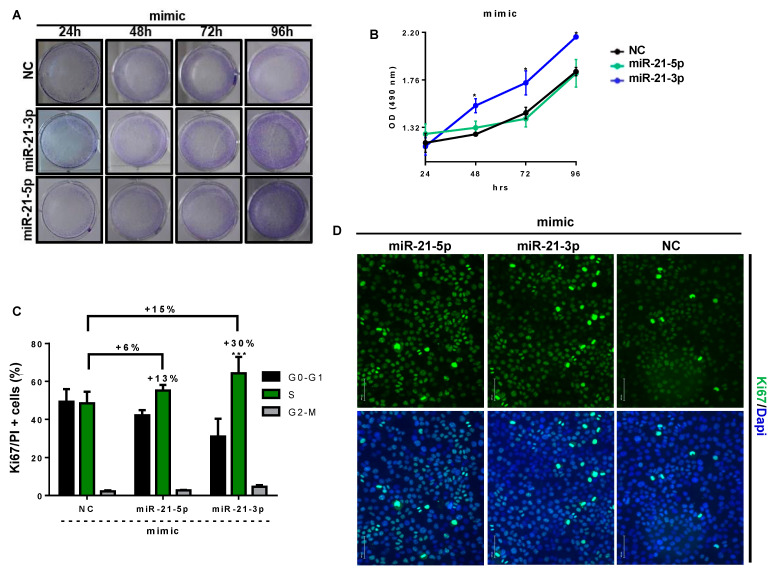
miR-21-3p gain of function promotes KC proliferation. (**A**) Crystal violet staining at different time pointa (24 h, 48 h, 72 h, and 96 h) of HaCaT cells transfected with negative control (NC) or miR-21-5p or miR-21-3p mimics. (**B**) XTT assays at different time points (24 h, 48 h, 72 h, and 96 h) of HaCaT cells transfected with negative controls (NC), miR-21-5p or miR-21-3p mimics. (**C**) Cell cycle analysis with double Ki-67/PI staining 24 h post-transfection of HaCaT cells with negative control (NC) or miR-21-5p or miR-21-3p mimics. (**D**) Ki67 immunostaining 24 h post transfection of HaCaT cells with negative control (NC) or miR-21-5p or miR-21-3p mimics. Data information: results are presented as mean values ± SEM. Data are representative of three independent experiments performed in triplicate. The statistical comparison between groups was performed by using the two-way ANOVA test: * *p* < 0.05, *** *p* < 0.001.

**Table 1 cells-10-02547-t001:** TaqMan primers.

Primers	Assay ID	Catalog Number
miR-21-5p	Hsa-miR-21-5p	477,975
miR-21-3p	Hsa-miR-21-3p	477,973
miR-21-3p	Rno-miR-21-3p	Rno480,993
miR-16-5p	Hsa-miR-16-5p	477,860
miR-16-5p	Rno-miR-16-5p	Rno481,312

**Table 2 cells-10-02547-t002:** Comparative table with GO terms comparing the data from Pasquali et al., 2019 [9] and this work. Red: upregulated DEGs; blue: downregulated DEGs.

	Pasquali et al. PP-H					This Work					
Term	Overlap	*p*-Value	Adjusted *p*-Value	Z-Score	Enrichment	Count	Pop Hits	*p*-Value	Benjiamini	FDR %	Enrichment
DNA-dependent DNA replication (GO:0006261)	40/118	1.56 × 10^−15^	1.21 × 10^−11^	−3.19	GO-BP-UP	5	18	0.014144	0.28673763	22.8965119	DAVID-UP
DNA replication (GO:0006260)	37/125	2.01 × 10^−12^	3.9 × 10^−9^	−3.16	GO-BP-UP	41	155	3.95 × 10^−17^	2.68 × 10^−14^	7.21 × 10^−14^	DAVID-UP
Double-strand break repair (GO:0006302)	42/164	1.32 × 10^−11^	2.04 × 10^−8^	−3.26	GO-BP-UP	14	66	4.39 × 10^−5^	0.00275283	0.08007114	DAVID-UP
G2/M transition of mitotic cell cycle (GO:0000086)	35/130	1.49 × 10^−10^	1.44 × 10^−7^	−3.13	GO-BP-UP	34	137	2.22 × 10^−13^	8.35 × 10^−11^	4.04 × 10^−10^	DAVID-UP
Viral DNA repair (GO:0046787)	38/160	1.28 × 10^−9^	8.27 × 10^−7^	−3.31	GO-BP-UP						
DNA repair (GO:0006281)	54/279	1.7 × 10^−9^	1.01 × 10^−6^	−3.99	GO-BP-UP	39	235	1.23 × 10^−7^	3.2 × 10^−7^	2.24 × 10^−6^	DAVID-UP
Keratinocyte development (GO: 0003334)	16/59	1.29 × 10^−5^	8.56 × 10^−4^	−2.32	GO-BP-UP						
Keratinocyte differentiation (GO:0030216)	19/80	1.68 × 10^−5^	1.03 × 10^−3^	−2.56	GO-BP-UP						
NIK/NF-κB signaling (GO:0038061)	16/64	3.9 × 10^−5^	2.19 × 10^−3^	−2.28	GO-BP-UP	18	66	5.1 × 10^−8^	8.65 × 10^−6^	9.31 × 10^−5^	DAVID-UP
Type I interferon signaling pathway (GO:0060337)	27/148	5.68 × 10^−5^	3 × 10^−3^	−2.95	GO-BP-UP						
Limb spinous cell differentiation (GO:0060890)	18/59	1.88 × 10^−12^	6.87 × 10^−10^	−2.66	GO-BP-Down						
Limb granular cell differentiation (GO:0060891)	18/59	1.88 × 10^−12^	6.87 × 10^−10^	−2.65	GO-BP-Down						
Keratinocyte development (GO:0003334)	18/59	1.88 × 10^−12^	6.87 × 10^−10^	−2.63	GO-BP-Down						
Keratinocyte differentiation (GO:0030216)	20/80	6.78 × 10^−12^	1.2 × 10^−9^	−2.8	GO-BP-Down						
Skin epidermis development (GO:0098773)	18/94	7.82 × 10^−9^	1.18 × 10^−6^	−2.84	GO-BP-Down						
Epidermis development (GO:0008584)	14/68	1.39 × 10^−7^	2.05 × 10^−5^	−2.49	GO-BP-Down						
Epidermis morphogenesis (GO:0048730)	14/75	5 × 10^−7^	7.18 × 10^−5^	−2.54	GO-BP-Down						
Epidermis cell differentiation (GO:0009913)	17/124	2.91 × 10^−6^	4.08 × 10^−4^	−2.98	GO-BP-Down						
Peptide cross-linking (GO:0018149)	17/128	4.52 × 10^−6^	6.19 × 10^−4^	−3.22	GO-BP-Down						
Fat cells differentiation (GO:0045444)	Sep-44	2.57 × 10^−5^	3.36 × 10^−3^	−2.64	GO-BP-Down						

## Data Availability

GEO accession: GSE154968/ Title: A global transcriptome analysis of human.

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
