# Peer review of "miR-21-3p/IL-22 Axes Are Major Drivers of Psoriasis Pathogenesis by Modulating Keratinocytes Proliferation-Survival Balance and Inflammatory Response"

_cells, 2021, doi:10.3390/cells10102547_

Round 1
Reviewer 1 Report
In this manuscript by Abdallah et al, the authors explored the roles of miR-21-3p, the passenger strand of miR-21, in psoriasis. They show that miR-21-5p and 3p are induced by IL-22 in keratinocytes via NF-kB and STAT3, that IL-22 and miR-21-3p are positively correlated in mouse psoriasis model as well as human skin biopsies, identify the transcriptional changes upon miR-32-3p overexpression and show that miR-21-3p promotes keratinocyte proliferation and migration.
This is a very comprehensive study on miR-21-3p in psoriasis and complements previous studies on miR-21-5p. Experimental details are missing at some places and need more explanation.
Specific comments:
1) Introduction: while IL-22 has unequivocally a role in epidermal hyperprliferation in psoriasis, and it contributes to the pathogenesis, the IL-23/IL-17 axis is more central in the pathogenesis. This should be mentioned.
“least thermodynamic stable strand”- the leader strand is the more stable strand, please correct.
2) Figure 1C - while miR-21-3p is indeed induced in the model, the baseline level is very low (if the axes are comparable). Whether this is biologically significant could only be decided if 3p could be specifically inhibited. Although low levels of the miRNA can also be functional this should be discussed in the discussion section.
3) Figure 4 C: STAT3 is not secreted from the cells. The appropriate way to investigate STAT3 and pSTAT3 is e.g. by western blot, not by ELISA as stated in the figure legend. Either a Western blot should be added or the panel should be skipped.
4) Figure 4 D: This panel is very unclear. Was miR-21-3p or 5p analyzed? By which method, qPCR? This is not stated in the legend. Moreover, was the efficacy of the siRNAs controlled in any way? In other words, were p65 and STAT3 as well as their phosphorylated forms decreased in siRNA treated cells? A western blot could answer this question.
5) Figure 6: how was miR-21-3p overexpressed? Transiently by mimics? In what concentration? After how long time were cells harvested? Experimental details are missing.
Moreover, Figure 6 Is hardly visible. I suggest to move this to a supplementary figure and then each part could be a whole page for better readability.
6) Figure 7: instead of just assessing the number of overlapping genes , the effect of miR-21-3p vs. the keratinocyte transcriptome in psoriasis could be better assessed by enrichment analysis (GSEA). i.e. were the genes induced by miR-21-3p enriched among the genes upregulated in PP keratinocytes?
7) Figure 8 the inhibitor results are not very convincing, perhaps these could be moved to supplementary material? Have the authors attempted to quantify the Ki67 staining (positive cells/field of view)?
Author Response
Please see attachement

Reviewer 2 Report
General comments:
The study of cutaneous miRNAs is a field that requires further understanding. This paper focuses on the study of miR-21-3p function in skin biology and psoriasis. The results presented are interesting and cover various important aspects, notably by joining in vivo and in vitro experimentation as well as the study of psoriatic skin biopsies. However, the manuscript presents a significant amount of results that ultimately make it difficult to read and interpret. This paper would benefit from being shortened by further sorting the results and highlighting the most promising ones, as well as putting the validations into supplementary materials. Moreover, it would be interesting if the authors could be more concise in their writing of the results section and discuss more about the impact of their study.
Comments:
- Please detail the choice of the methods: why utilize mice IL22-/- 3 to study the correlation between miR-21 and IL22, but also carry out correlation tests for NFkappaB and STAT3 on keratinocytes lines instead of IL22-/- mice?
- Why the concentration of 50 ng/ml has been chosen for IL-22 treatment of HaCaT and NHEK cells? A dose response would have been more informative.
- It is not totally fair to conclude that miR-21-3p/ILL-22 axes drive psoriasis pathogenesis, while no correlation has been established between miR-21-3p and specific disease markers (such as IL-17, psoriasin, elafin) while using the psoriatic model.
- The relevance of wound healing testing is not clear in this article, since the aim of the study was to evaluate the role of miR-21-3p in psoriasis? Although these results are interesting, their relevance is not obvious to me in this article.
- How the authors conclude that miR-21-3p modulate innate and adaptative immunity? No results shown have been tested on immune cells.
Minor comments:
- Supplementary materials: please do not forget to include the figures in the supplementary materials of the final version.
- Please number the sections and sub-sections.
- Figure 1 : In the figure legend, it is indicated that the scale is from 0 (absence of severity) to 4 (highest severity), but there is some higher score in the graph.
- Line106: IL22-/- mice
- RT-qPCR : it is not clear why the relative expression is reported on miR-16-5p. Please specify.
- In Figure 1 B-C, the differences between the two set of points for WT and IL22-/- are not clear. Please identify that they are healthy vs psoriatic mice.
- Figure 2B: the name of the Y axis seems wrong? miR-21-3p?
- Figure 2C and D: is it possible to distinguish data from biopsies of healthy skin from that of psoriatic skin?
- Figure 3A: Please indicate what the green dots on days 4 and 8 represent.
- Respect gene nomenclature through the manuscript (italic).
- Lines 271-289: to lighten the results section, please transfer these lines in the discussion section.
- Figure 7 and the results associated: I propose to transfer these results in the supplementary material but discuss them in the discussion section.
- The post-hoc tests are not specified (neither in the figure legends nor in the statistics section). Why present data as mean ± SEM and not ± SD?
- Results section 4-5-6: Why discuss supplementary data in the results section while the results are not presented directly in the manuscript? It can be confusing and it makes the comprehension of the article difficult.
- In the results and discussion, please put the references only at the end of the sentence.
- Line 299: Moreover, whatever no matter the number of DEGs in common, the significant enriched biological processes of total up-regulated genes in KCs overexpressing miR-21-3p….
Author Response
Please see attachement

Reviewer 3 Report
In the manuscript entitled, “miR-21-3p/IL-22 axes are major drivers of psoriasis pathogenesis by modulating keratinocytes proliferation-survival balance and inflammatory response” by Abdallah et al., the authors have studied the role of IL-22-dependent regulation of the microRNA, miR-21, in the development of psoriasis pathogenesis. The authors have utilized the IMQ-driven psoriasis model. While the expression of the passenger miR-21-3p strand was correlated with IL-22 expression in a STAT3 and NF-kB-dependent manner, the correlation to miR-21-5p was marginal. The IL-22-induced miR-21-3p was further validated functionally using the RNAi - Inducible Luciferase Expression System. The authors have subsequently identified genes involved in keratinocyte hyperproliferation which could be the plausible underlying cause of development of psoriatic lesions. While the authors have demonstrated the role of IL-22 in miR-21-3p-mediated psoriasis development, this work could be improved by clarifying more concerns.
- For gene expression analysis, it is recommended to normalize the control value to 1 in all the graphs for easy visualization and better understanding of readers.
- In Fig 2C and 2D, the value of Pearson correlation coefficient is not reported.
- In Fig3C, day Post IMQ barplot, is the comparison between IMQ vs IL-22+IMQ statistically significant?
- In Fig 4D, the authors should show the individual expression levels of miR-21-5p and miR-21-3p.
- Can the authors quantify the crystal violet staining data in Fig 8A? It seems that the treatment of HaCaT cells with miR-21-5p shows an increase in crystal violet staining which is in contrast with other subsequent assays. Can the authors explain these differences?
- In Fig8C, NC group in mimic and inhibitor-treated cells have a huge difference in number of positive cells between these controls. Can the authors justify these results?
- In Fig 3 graph axis, vehicle is spelled wrong.
- The authors have not provided the supplementary data for the manuscript. Although the results presented in the main manuscript are in line with the proposed hypothesis, a conclusion could not be drawn in absence of the data.
Author Response
Please see attachement

Round 2
Reviewer 3 Report
The authors have addressed the concerns raised earlier. The article may be accepted in the current form.